# Recognizing retinal ganglion cells in the dark

**Emile Richard**
Stanford University
emileric@stanford.edu

**Georges Goetz**
Stanford University
ggoetz@stanford.edu

**E.J. Chichilnisky**
Stanford University
ej@stanford.edu

## Abstract

Many neural circuits are composed of numerous distinct cell types that perform different operations on their inputs, and send their outputs to distinct targets. Therefore, a key step in understanding neural systems is to reliably distinguish cell types. An important example is the retina, for which present-day techniques for identifying cell types are accurate, but very labor-intensive. Here, we develop automated classifiers for functional identification of retinal ganglion cells, the output neurons of the retina, based solely on recorded voltage patterns on a large scale array. We use per-cell classifiers based on features extracted from electrophysiological images (spatiotemporal voltage waveforms) and interspike intervals (autocorrelations). These classifiers achieve high performance in distinguishing between the major ganglion cell classes of the primate retina, but fail in achieving the same accuracy in predicting cell polarities (ON vs. OFF). We then show how to use indicators of functional coupling within populations of ganglion cells (cross-correlation) to infer cell polarities with a matrix completion algorithm. This can result in accurate, fully automated methods for cell type classification.

## 1 Introduction

In the primate and human retina, roughly 20 distinct classes of retinal ganglion cells (RGCs) send distinct visual information to diverse targets in the brain [18, 7, 6]. Two complementary methods for identification of these RGC types have been pursued extensively. Anatomical studies have relied on indicators such as dendritic field size and shape, and stratification patterns in synaptic connections [8] to distinguish between cell classes. Functional studies have leveraged differences in responses to stimulation with a variety of visual stimuli [9, 3] for the same purpose. Although successful, these methods are difficult, time-consuming and require significant expertise. Thus, they are not suitable for large-scale, automated analysis of existing large-scale physiological recording data. Furthermore, in some clinical settings, they are entirely inapplicable. At least two specific scientific and engineering goals demand the development of efficient methods for cell type identification:

- *Discovery of new cell types.* While ∼20 morphologically distinct RGC types exist, only 7 have been characterized functionally. Automated means of detecting unknown cell types in electrophysiological recordings would make it possible to process massive amounts of existing large-scale physiological data that would take too long to analyze manually, in order to search for the poorly understood RGC types.

- *Developing brain-machine interfaces of the future.* In blind patients suffering from retinal degeneration, RGCs no longer respond to light. Advanced retinal prostheses previously demonstrated *ex-vivo* aim at electrically restoring the correct neural code in each RGC type in a diseased retina [11], which requires cell type identification without information about the light response properties of RGCs.

In the present paper, we introduce two novel and efficient computational methods for cell type identification in a neural circuit, using spatiotemporal voltage signals produced by spiking cells

recorded with a high-density, large-scale electrode array [14]. We describe the data we used for our study in Section 2, and we show how the raw descriptors used by our classifiers are extracted from voltage recordings of a primate retina. We then introduce a classifier that leverages both hand-specified and random-projection based features of the electrical signatures of unique RGCs, as well as large unlabeled data sets, to identify cell types (Section 3). We evaluate its performance for distinguishing between midget, parasol and small bistratified cells on manually annotated datasets. Then, in Section 4, we show how matrix completion techniques can be used to identify populations of unique cell types, and assess the accuracy of our algorithm by predicting the polarity (ON or OFF) of RGCs on datasets where a ground truth is available. Section 5 is devoted to numerical experiments that we designed to test our modeling choices. Finally, we discuss future work in Section 6.

## 2    Extracting descriptors from electrical recordings

In this section, we define the electrical signatures that we will use in cell classification, and the algorithms that allow us to perform the statistical inference of cell type are described in the subsequent sections.

We exploit three electrical signatures of recorded neurons that are well measured in large-scale, high-density recordings. First, the electrical image (EI) of each cell, which is the average spatiotemporal pattern of voltage measured across the entire electrode array during the spiking of a cell. This measure provides information about the geometric and electrical conduction properties of the cell itself. Second, the inter-spike interval distribution (ISI), which summarizes the temporal separation between spikes emitted by the cell. This measure reflects the specific ion channels in the cell and their distribution across the cell. Third, the cross-correlation function (CCF) of firing between cells. This measure captures the degree and polarity of interactions between cells in generation of a spike.

### 2.1    Electrophysiological image calculation, alignment and filtering

The raw data we used for our numerical experiments consist of extracellular voltage recordings of the electrical activity of retinas from male and female macaque monkeys, which were sampled and digitized at 20 $k$Hz per channel over 512 channels laid out in a 60 $\mu$m hexagonal lattice (See Appendix for a 100 $m$s sample movie of an electrical recording). The emission of an action potential by a spiking neuron causes transient voltage fluctuations along its anatomical features (soma, dendritic tree, axon). By bringing an extracellular matrix of electrodes in contact with neural tissue, we capture the 2D projection of these voltage changes onto the plane of the recording electrodes (see Figure 1). With such dense multielectrode arrays, the voltage activity from a single cell is usually picked up on multiple electrodes. While the literature refers to this footprint as the *electrophysiological* or *electrical image* (EI) of the cell [13], it is an inherently spatiotemporal characteristic of the neuron, due to the transient nature of action potentials. In essence, it is a short movie ($\sim 1.5$ $m$s) of the average electrical activity over the array during the emission of an action potential by a spiking neuron, which can include the properties of other cells whose firing is correlated with this neuron.

We calculated the electrical images of each identified RGC in the recording as described in the literature [13]. In a 30–60 minute recording, we typically detected 1,000–100,000 action potentials per RGC. For each cell, we averaged the voltages recorded over the entire array in a 1.5 $m$s window starting .25 $m$s before the peak negative voltage sample for each action potential. We cropped from the electrode array the subset of electrodes that falls within a 125 $\mu$m radius around the RGC soma (see Figure 1) in order to represent each EI by a $30 \times 19$ matrix (time points $\times$ number of electrodes in a 125 $\mu$m radius), or equivalently a 570 dimensional vector. We augment the training data by exploiting the symmetries of the (approximately) hexagonal grid of the electrode array. We form the training data EIs from original EIs, rotating them by $i\pi/3, i = 1, \cdots, 6$ and the reflection of each (12 spatial symmetries in total). The characteristic radius (125 $\mu$m here) used to select the central portion of the EI is a hyper-parameter of our method which controls the signal to noise ratio in the input data (see Section 5, Figure 3 middle panel).

In the Appendix of this paper we describe 3 families (subdivided into 7 sub-families) of filters we manually built to capture anatomical features of the cell. In particular, we included filters corresponding to various action potential propagation velocities at level of the the axon and hard-coded a parameter which captures the soma size. These quantities are believed to be indicative of cell type.

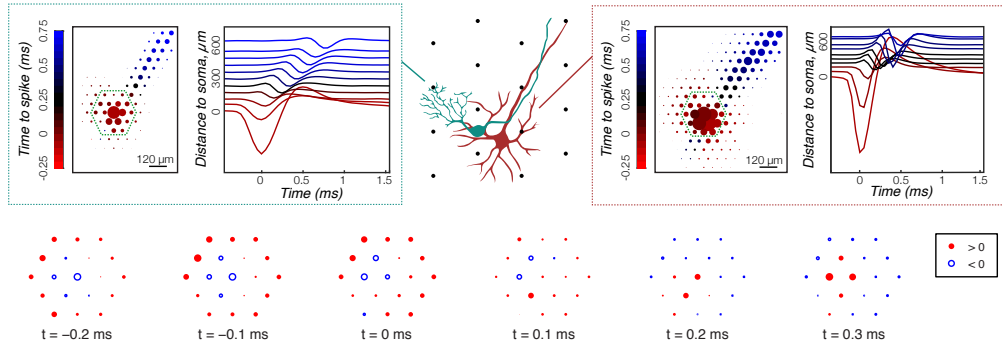

Figure 1: EIs and cell morphology. *(Top row)* Multielectrode arrays record a 2D projection of spatio-temporal action potentials, schematically illustrated here for a midget (left) and a parasol (right) RGC. Midget cells have an asymmetric dendritic field, while parasol cells are more isotropic. *(Bottom row)* Temporal evolution of the voltage recorded on the electrodes located within a 125 $\mu$m-radius around the electrode where the largest action potential was detected, which we use for cell type classification. Amplitude of the circles materialize signal amplitude. Red circles — positive voltages, blue circles — negative voltages.

We filtered the spatiotemporally aligned RGC electrical images with our hand defined filters to create a first feature set. In separate experiments we also filtered aligned EIs with iid Gaussian random filters (as many as our features) in the fashion of [17], see Table 1 to compare performances.

## 2.2 Interspike Intervals

The statistics of the timing of action potential trains are another source of information about functional RGC types. Interspike intervals (ISIs) are an estimate of the probability of emission of two consecutive action potentials within a given time difference by a spiking neuron. We build histograms of the times elapsed between two consecutive action potentials for each cell to form its ISI. We estimate the interspike intervals over 100 $ms$, with a time granularity of 0.5 $ms$, resulting in 200 dimensional ISI vectors. ISIs always begin by a refractory period (i.e. a duration over which no action potentials occur, following a first action potential). This period lasts the first 1-2 $m$s. ISIs then increase before decaying back to zero at rates representative of the functional cell type (see Figure 2, left hand side). We describe each ISI using the values of time differences $\Delta t$ where the smoothed ISI reaches $20, 40, 60, 80, 100\%$ of its maximum value as well as the slopes of the linear interpolations between each consecutive pair of points.

## 2.3 Cross-correlation functions and electrical coupling of cells

There is in the retina a high probability of joint emission of action potentials between neighboring ganglion cells of the same type, while RGCs of antagonistic polarities (ON vs OFF cells) tend to exhibit strongly negatively correlated firing patterns [16, 10]. In other words, the emission of an action potential in the ON pathway leads to a reduced probability of observing an action potential in the OFF pathway at the same time. The cross-correlation function of two RGCs characterizes the probability of joint emission of action potentials for this pair of cells with a given latency, and as such holds information about functional coupling between the two cells. Cross-correlations between different functional RGC types have been studied extensively in the literature previously, for example in [10]. Construction of CCFs follows the same steps as ISI computation: we obtain the CCF of pairs of cells by building histograms of time differences between their consecutive firing times. A large CCF value near the origin is indicative of positive functional coupling whereas negative coupling corresponds to a negative CCF at the origin (see Figure 2, the three panels on the right).

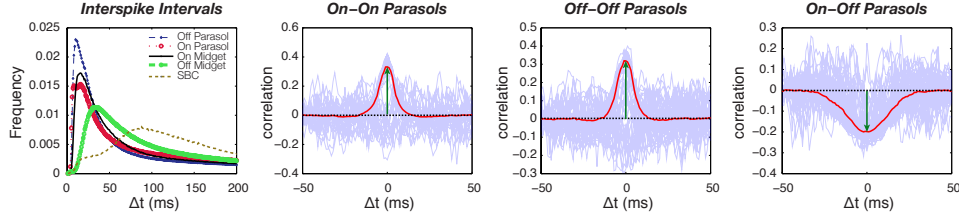

Figure 2: *(Left panel)* Interspike Intervals for the 5 major RGC types of the primate retina. *(Right panels)* Cross-correlation functions between parasol cells. Purple traces: single pairwise CCF. Red line: population average. Green arrow: strength of the correlation.

## 3 Learning electrical signatures of retinal ganglion cells

### 3.1 Learning dictionaries from slices of unlabeled data

Learning descriptors from unlabeled data, or dictionary learning [15], has been successfully used for classification tasks in high-dimensional data such as images, speech and texts [15, 4]. The methodology we used for learning discriminative features given a relatively large amount of unlabeled data follows closely the steps described in [4, 5].

**Extracting independent slices from the data.**   The first step in our approach consists of extracting independent (as much as possible) *slices* from data points. One can think of a slice as a subset of the descriptors that is (nearly) independent from other subsets. In image processing the analogue object is named a *patch*, i.e. a small sub-image. In our case, we used 8 data slices. The ISI descriptors form one such slice, the others are extracted from EIs. It is reasonable to assume ISI features and EI descriptors are independent quantities. After aligning the EIs and filtering them with a collection of 7 filter banks (see Appendix for a description of our biologically motivated filters), we group each set of filtered EIs. Each group of filters reacts to specific patterns in EIs: rotational motion driven by dendrites, radial propagation of the electrical signal along the axon and the direction of propagation constitute behaviors captured by distinct filter banks. Thereby, we treat the response of data to each one of them as a unique data slice. Each slice is then whitened [5], and finally we perform sparse $k$-means on each slice separately, $k$ denotes an integer which is a parameter of our algorithm. That is, letting $\mathbf{X} \in \mathbb{R}^{n \times p}$ denote a slice of data ($n$: number of data points and $p$: dimensionality of the slice) and $\mathcal{C}_{n,k}$ denote the set of cluster assignment matrices: $\mathcal{C}_{n,k} = \{\mathbf{U} \in \{0,1\}^{n \times k} \ : \ \forall i \in [n] \,, \ \|\mathbf{U}_{i,\cdot}\|_0 = 1\}$, we consider the optimization problem

$$\min \|\mathbf{X} - \mathbf{U}\mathbf{V}^\mathsf{T}\|_\mathsf{F}^2 + \eta\|\mathbf{V}\|_1 \quad \text{s.th.} \quad \mathbf{U} \in \mathcal{C}_{n,k} \ , \ \mathbf{V} \in \mathbb{R}^{p \times k} \ . \tag{1}$$

**Warm-starting $k$-means with warm started NMF.**   In order to solve the optimization problem (1), we propose a coarse-to-fine strategy that consists in relaxing the constraint $\mathbf{U} \in \mathcal{C}_{n,k}$ in two steps. We initially relax the constraint $\mathbf{U} \in \mathcal{C}_{n,k}$ completely and set $\eta = 0$. That is, we consider problem (1) where we substitute $\mathcal{C}_{n,k}$ with the larger set $\mathbb{R}^{n \times k}$ and run an alternate minimization for a few steps. Then, we replace the clustering constraint $\mathcal{C}_{n,k}$ with a nonnegativity constraint $\mathbf{U} \in \mathbb{R}_+^{n \times k}$ while retaining $\eta = 0$. After a few steps of nonnegative alternate minimization we activate the constraint $\mathbf{U} \in \mathcal{C}_{n,k}$ and finally raise the value of $\eta$. This warm-start strategy systematically resulted in lower values of the objective compared to random or $k$-means++ [1] initializations.

### 3.2 Building feature vectors for labeled data

In order to extract feature vectors from labeled data we first extract slice each data point: we extract ISI features on the one hand and filter each data point with all filter families. Each slice is separately whitened and compared to the cluster centers of its slice. For this, we use the matrices $\mathbf{V}^{(j)}$ of cluster centroids computed for the all slices $j = 1, \cdots, 8$. Letting $\mathsf{s}(\cdot, \kappa)$ denote the soft thresholding operator $\mathsf{s}(\mathbf{x}, \kappa) = (\text{sign}(\mathbf{x}_i) \max\{|\mathbf{x}_i| - \kappa, 0\})_i$, we compute $\tilde{\mathbf{x}}^{(j)} = \mathsf{s}(\mathbf{V}^{(j)\mathsf{T}}\mathbf{x}^{(j)}, \kappa)$ for each slice, which is the soft-thresholded inner products of the corresponding slice of the data point $\mathbf{x}^{(j)}$ with all cluster centroids for the same slice $j$. We concatenate the $\tilde{\mathbf{x}}^{(j)}$s from different slices and use

the resulting encoded point to predict cell types: $\tilde{\mathbf{x}} = (\tilde{\mathbf{x}}^{(j)})_j$. The last step is performed either by feeding concatenated vectors $\tilde{\mathbf{x}}$ together with the corresponding label to a logistic regression classifier which handles multiple classes in a one-versus-all fashion, or to a random forest classifier.

## 4    Predicting cell polarities by completing the RGC coupling matrix

We additionally exploit pairwise spike train cross-correlations to infer RGC polarities (ON vs OFF) and estimate the polarity vector $\mathbf{y}$ by using a measure of the pairwise functional coupling strength between cells. The rationale behind this approach is that neighboring cells of the same polarity will tend to exhibit positive correlations between their action potential spike trains, corresponding to positive functional coupling. If the cells are of antagonistic polarities, functional coupling strength will be negative. The coupling of two neighboring cells $i, j$ can therefore be modeled as $\mathbf{c}_{\{i,j\}} \simeq \mathbf{y}_i \mathbf{y}_j$, where $\mathbf{y}_i, \mathbf{y}_j \in \{+1, -1\}$ denote cell polarities. Because far apart cells do not excite or inhibit each other, to avoid incorporating noise in our model, we choose to only include estimates of functional coupling strengths between neighboring cells. The neighborhood size is a hyper-parameter of this approach that we study in Section 5.

If $\mathcal{G}$ denotes the graph of neighboring cells in a recording, we only use cross-correlations for spike trains of cells which are connected with an edge in $\mathcal{G}$. Since we can estimate the position of each RGC in the lattice from its EI [13], we therefore can form the graph $\mathcal{G}$, which is a 2-dimensional regular geometric graph. If $q$ is the number of edges in $\mathcal{G}$, let $\mathcal{P}$ denote the linear map $\mathbb{R}^{n \times n} \to \mathbb{R}^q$ returning the values $\mathcal{P}(\mathbf{C}) = (\mathbf{C}_{i,j})_{\{i,j\} \in E(\mathcal{G})}$ for cells $i$ and $j$ located within a critical distance. We use $\mathcal{P}^*$ to denote the adjoint (transpose) operator. The complete matrix of pairwise couplings can then be written — up to observation noise — as $\mathbf{y}\mathbf{y}^{\mathsf{T}}$, where $\mathbf{y} \in \{+1, -1\}^n$ is the vector of cell polarities ($+1$ for ON and $-1$ for OFF cells). Therefore, the observation can be modeled as:

$$\mathbf{c} = \mathcal{P}(\mathbf{y}\mathbf{y}^{\mathsf{T}}) + \varepsilon \quad \text{with} \quad \varepsilon \quad \text{observation noise.} \tag{2}$$

and the recovery of $\mathbf{y}\mathbf{y}^{\mathsf{T}}$ is then formulated as a standard matrix completion problem.

### 4.1    Minimizing the nonconvex loss using warm-started Newton steps

In this section, we show how to estimate $\mathbf{y}$ given the observation of $\mathbf{c} = \mathcal{P}(\mathbf{y}\mathbf{y}^{\mathsf{T}}) + \varepsilon$ by minimizing the non-convex loss $\ell(\mathbf{z}) = \frac{1}{2}\|\mathcal{P}(\mathbf{z}\mathbf{z}^{\mathsf{T}}) - \mathbf{c}\|_2^2$. Even though minimizing this degree 4 polynomial loss function is NP-hard in general, we propose a Newton method warm-started with a spectral heuristic for approaching the solution (see Algorithm 1). In similar contexts, when the sampling of entries is uniform, this type of spectral initialization followed by alternate minimization has been proven to converge to the global minimum of a least-squared loss, analogous to $\ell$ [12].

While our sampling graph $\mathcal{G}$ is not an Erdos-Renyi graph, we empirically observed that its regular structure enables us to come up with a reliable initial spectral guess that falls within the basin of attraction of the global minimum of $\ell$. In the subsequent Newton scheme, we iterate using the shifted Hessian matrix $\mathbf{H}(\mathbf{z}) = \mathcal{P}^*\left(2\,\mathcal{P}(\mathbf{z}\mathbf{z}^{\mathsf{T}}) - \mathbf{c}\right) + \nu I_n$ where $\nu > 0$ ensures positive definiteness $\mathbf{H}(\mathbf{z}) \succ 0$. Whenever computing $\nu$ and $\mathbf{H}(\mathbf{z})^{-1}$ is expensive due to a potentially large number of cells $n$, then replacing $\mathbf{H}(\mathbf{z})^{-1}$ by a diagonal or scalar approximation $\alpha/\|\mathbf{z}\|_2^2$ reduces per iteration cost while resulting in a slower convergence. We refer to this method as a first-order method for minimizing the nonconvex objective, while ISTA [2] is a first order method applied to the convex relaxation of the problem as presented in the Appendix (see Figure 4, middle panel). Using the same convex relaxation we prove in the Appendix that the proposed estimator has a classification accuracy of at least $1 - b\|\varepsilon\|_\infty^2$ with $b \approx 2.91$.

---

**Algorithm 1** Polarity matrix completion

---

**Require:** $\mathbf{c}$ observed couplings, $\mathcal{P}$ the projection operator
    Let $\lambda, \mathbf{v}$ be the leading eigenpair of $\mathcal{P}^*(\mathbf{c})$
    Initialize $\mathbf{z}_0 \leftarrow n\sqrt{\lambda}\, \mathbf{v}/\sqrt{|\#\text{revealed entries}|}$
    **for** $t = 0, 1, \cdots$ **do**
        $\mathbf{z}_{t+1} \leftarrow \mathbf{z}_t - \mathbf{H}^{-1}(\mathbf{z}_t)\mathcal{P}^*\left(\mathcal{P}(\mathbf{z}_t\mathbf{z}_t^{\mathsf{T}}) - \mathbf{c}\right)\mathbf{z}_t$     $\backslash\backslash$    $\mathbf{H}(\mathbf{z}_t)$ is the Hessian or an approximation
    **end for**

---

| Input | EI & ISI | EI & ISI | EI & ISI | EI only | | |
| Task | our filters $k=30$ | rand. filters $k=50$ | rand. filters $k=10$ | our filters $k=30$ | ISI only | CCF |
| --- | --- | --- | --- | --- | --- | --- |
| T | **93.5** % (1.1 ) | 88.3 % (1.9) | **93.1** % (1.3) | 86.0 % (2.6) | 80.6 % (2.6) | – |
| P | **81.5** % (3.0) | **80.0** % (2.3) | 77.8 % (2.3) | 64.1 % (3.7) | 76.8 % (3.8) | 75.7 % (4.9) |
| T+P | **78.0** % ( 3.3) | 66.7 % (1.9) | 72.0 % (1.7) | 60.4 % (2.9) | 64.7 % (2.9) | – |

Table 1: Comparing performance for input data sources and filters. T: cell type identification. P: polarity identification. T+P: cell type and polarity identification. EIs cropped within 125 $\mu$m from the central electrode.

# 5    Numerical experiments

In this section, we benchmark the performance of the cell type classifiers introduced previously on datasets where the ground truth was available. For the RGCs in those datasets, experts manually hand-labeled the light response properties of the cells in the manner previously described in the literature [9, 3]. Our unlabeled data contained 17,457 $\times$ 12 (spatial symmetries) data points. The labeled data consists of 436 OFF midget, 652 OFF parasol, 964 ON midget, 607 ON parasol and 169 small bistratified cells assembled from 10 distinct recordings.

**RGC classification from their electrical features.**    Our numerical experiment consists in hiding one out of 10 labeled recordings, learning cell classifiers on the 9 others and testing the classifier on the hidden recording. We chose to test the performance of the classifier against individual recordings for two reasons. Firstly, we wanted to compare the polarity prediction accuracy from electrical features with the prediction made by matrix completion (see Section 4) and the matrix completion algorithm takes as input pairwise data obtained from a single recording only. Secondly, experimental parameters likely to influence the EIs and ISIs such as recording temperature vary from recording to recording, but remain consistent within a recording. Since we want the reported scores to reflect expected performance against new recordings, not including points from the test distribution gives us a more realistic proxy to the true test error.

In Table 1 we report classification accuracies on 3 different classification tasks:

1. Cell type identification (T): midget vs. parasol vs. small bistratified cells;
2. Polarity identification (P): ON versus OFF cells;
3. Cell type and polarity (T+P): ON-midget vs. ON-parasol vs. OFF-midget vs. OFF-parasol vs. small bistratified.

Each row of the table contains the data used as input. The first column represents the results for the method where the dictionary learning step is performed with $k=30$, and EIs are recorded within a radius of 125 $\mu$m from the central electrode (19 electrodes on our array). We compare our method with an identical method where we replaced the hand-specified filters by the random Gaussian filters of [17] (second column for $k=50$ and third for $k=10$). The performance of random filters opens perspectives for learning deeper predictors using random filters in the first layer. The impact of $k$ on our filters can be seen in Figure 3, left-hand panel: larger $k$ seems to bring further information for polarity prediction but not for cell type classification, which leads to an optimal choice $k \simeq 20$ in the 5-class problem. In the 4th and 5th columns, we used only part of the features sets at our disposal, EIs only and ISIs only respectively. These results confirm that the joint use of both EIs and ISIs for cell classification is beneficial. Globally, cell type identification turns out to be an easier task than polarity prediction using per cell descriptors.

Figure 3 middle panel illustrates the impact of EI diameter on classification accuracy. While a larger recording radius lets us make use of more signal, the amount of noise incorporated also increases with the number of electrodes taken into account and we observe a trade-off in terms of signal to noise ratio on all three tasks. An interesting observation is the second jump in the accuracy of cell-type prediction around an EI diameter of $325\mu$m, at which point we attain a peak performance of $96.8\% \pm 1.0$. We believe this jump takes place when axonal signals start being incorporated in the EI, and we believe these signals to be a strong indicator of cell type because of

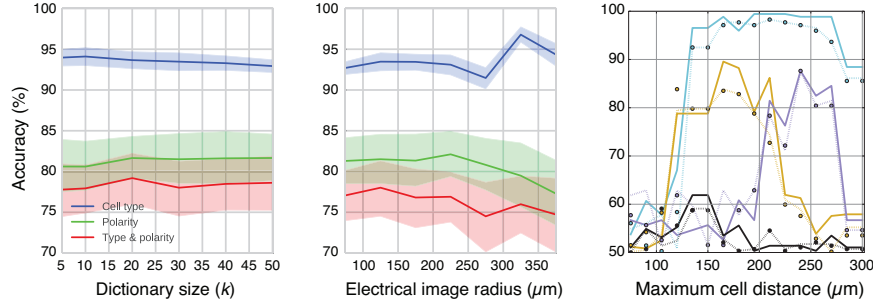

Figure 3: *(Left panel)* Effect of the dictionary size $k$ and *(Middle panel)* EIs radius on per cell classification. *(Right panel)* Effect of the neighborhood size on polarity prediction using matrix completion.

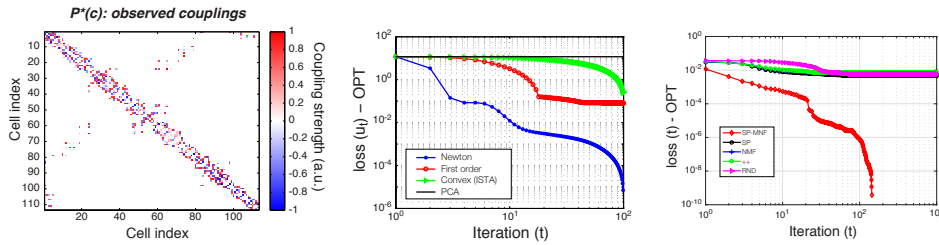

Figure 4: *(Left panel)* Observed coupling matrix. *(Middle panel)* Convergence of matrix completion algorithms. *(Right panel)* $k$-means with our initialization (SP-NMF) versus other choices.

known differences in axonal conduction velocities [13]. Prediction variance is also relatively low for cell-type prediction compared to polarity prediction, while predicting polarity turns out to be significantly easier on some datasets than others. On average, the logistic regression classifier we used performed slightly better ($\sim +1\%$) than random forests on the various tasks and data sets at our disposal.

**Matrix completion based polarity prediction.** Matrix completion resulted in $> 90\%$ accuracy on three out of 10 datasets and in an average of $66.8\%$ accuracy in the 7 other datasets. We report the average performance in Table 1 even though it is inferior to the simpler classification approach for two reasons: (a) the idea of using matrix completion for this task is new and (b) it has a high potential, as demonstrated by Figure 3, right hand panel. On some datasets, matrix completion has 100%accuracy. However, on other datasets, either because of issues a fragile spike-sorting, or of other noise, the approach does not do as well. In Figure 3 (right hand side) we examine the effect of the neighborhood size on prediction accuracy. Colors correspond to different datasets. For sake of readability, we only show the results for 4 out of 10 datasets: the best, the worse and 2 intermediary. The sensitivity to maximum cell distance is clear on this plot. Bold curves correspond to the prediction resulting after 100 steps of our Newton algorithm. Dashed curves correspond to predictions by the first order (nonconvex) method stopped after 100 steps, and dots are prediction accuracies of the leading singular vector, i.e. the spectral initialization of our algorithm. Overall, the Newton algorithm seems to perform better than its rivals, and there appears to be an optimal radius to choose for each dataset which corresponds to the characteristic distance between pairs of cells (here only Parasols). This parameter varies from dataset to dataset and hence requires parameter tuning before extracting CCF data in order to get the best performance out of the algorithm.

**Warm-start strategy for dictionary learning.** We refer to Figure 4, right hand panel for an illustration of our warm-start strategy for minimizing (1) as described in Section 3.1. There, we compare dense ($\eta = 0$) $k$-means initialized with our double-warm start (25 steps of unconstrained alternate minimization and 25 steps of nonnegative alternate minimization, referred to as SP-NMF), with a single spectral warm start 50 steps unconstrained alternate minimization initialization (SP) and a 50 steps nonnegative alternate minimization (NMF) as well as with two standard baselines which

are random initialization and $k$-means++ initializations [1]. We postpone theoretical study of this initialization choice to future work. Note that each step of the alternate minimization involves a few matrix-matrix products and element-wise operations on matrices. Using a `NVIDIA Tesla K40 GPU` drastically accelerated these steps, allowing us to scale up our experiments.

## 6  Discussion

We developed accurate cell-type classifiers using a unique collection of labeled and unlabeled electrical recordings and employing recent advances in several areas of machine learning. The results show strong empirical success of the methodology, which is highly scalable and adapted for major applications discussed below. Matrix completion for binary classification is novel, and the two heuristics we used for minimizing our non-convex objectives show convincing superiority to existing baselines. Future work will be dedicated to studying properties of these algorithms.

*Recording Methods.* Three major aspects of electrical recordings are critical for successful cell type identification from electrical signatures. First, high spatial resolution is required to detect the fine features of the EIs; much more widely spaced electrode arrays such as those often used in the cortex may not perform as well. Second, high temporal resolution is required to measure the ISI accurately; this suggests that optical measurements using Ca++ sensors would not be as useful as electrical measurements. Third, large-scale recordings are required to detect many pairs of cells and estimate their functional interactions; electrode arrays with fewer channels may not suffice. Thus, large-scale, high-density electrophysiological recordings are uniquely well suited to the task of identifying cell types.

*Future directions.* A probable source of variability in cell type classification is differences between retinal preparations, including eccentricity in the retina, inter-animal variability, and experimental variables such as temperature and signal-to-noise of the recording. In the present data, features were defined and assembled across a dozen different recordings. This motivates transfer learning to account for such variability, exploiting the fact that although the features may change somewhat between preparations (target domains), the underlying cell types and the fundamental differences in electrical signatures are expected to remain. We expect future work to result in models that enjoy higher complexity thanks to training on larger datasets, thus achieving invariance to ambient conditions (eccentricity and temperature) automatically. The model we used can be interpreted as a single-layer neural network. A straightforward development would be to increase the number of layers. The relative success of random filters on the first layer is a sign that one can hope to get further automated improvement by building richer representations from the data itself and with minimum incorporation of prior knowledge.

*Application.* Two major applications are envisioned. First, an extensive set of large-scale, high-density recordings from primate retina can now be mined for information on infrequently-recorded cell types. Manual identification of cell types using their light response properties is extremely labor-intensive, however, the present approach promises to facilitate automated mining. Second, the identification of cell types without light responses is fundamental for the development of high-resolution retinal prostheses of the future [11]. In such devices, it is necessary to identify which electrodes are capable of stimulating which cells, and drive spiking in RGCs according to their type in order to deliver a meaningful visual signal to the brain. For this futuristic brain-machine interface application, our results solve a fundamental problem. Finally, it is hoped that these applications in the retina will also be relevant for other brain areas, where identification of neural cell types and customized electrical stimulation for high-resolution neural implants may be equally important in the future.

## Acknowledgement

We are grateful to A. Montanari and D. Palanker for inspiring discussions and valuable comments, and C. Rhoades for labeling the data. ER acknowledges support from grants AFOSR/DARPA FA9550-12-1-0411 and FA9550-13-1-0036. We thank the Stanford Data Science Initiative for financial support and NVIDIA Corporation for the donation of the Tesla K40 GPU we used. Data collection was supported by National Eye Institute grants EY017992 and EY018003 (EJC). Please contact EJC (ej@stanford.edu) for access to the data.

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
