[Supplementary Material]

# Recognizing retinal ganglion cells in the dark (appendix)

**Emile Richard**
Stanford University
emileric@stanford.edu

**Georges Goetz**
Stanford University
ggoetz@stanford.edu

**EJ Chichilnisky**
Stanford University
ej@stanford.edu

## 1 EI and MEA movies

The files attached to this document contain short videos that are intended to help the reader get a better intuition on our data. The movies are:

1. `videoMEA.webm`. This is a visual representation of 100 $m$s of data recorded on our 512-channel multielectrode array grid. The file can be read using any modern web browser. Red indicates that at a given instant in time, positive voltages were recorded on a channel, and blue indicates that negative voltages were acquired. The movie lasts 1 minute, so it is slowed down 600 times as compared to real time.

2. `videoEI`. The folder contains `.gif` files of average EIs for different cell types and recordings. Each video therefore represents the average of all EIs for RGCs of a single type detected within a unique recording. Those RGCs were classified using their visible light response properties. An internal contest in our lab showed that humans can classify retinal ganglion cells barely better than random using unlabeled EIs. It is nonetheless possible to notice some differences between EIs of different cell types: one can somehow see that Parasols have bigger soma than Midgets, for instance.

## 2 Proposed EI filters

After alignment in space and time, EIs are filtered using 3 families of filters that reflect expected discriminative characteristics of the cells. Each filter family is written as the product of 3 terms: a first term comprises the propagation velocity of the signal and encodes the spatio-temporal dependency in the EI, and two other terms, one spatial $\psi(r)$ and one temporal $\psi(t)$ modulate the main term. In our numerical experiments the modulation envelope is encoded with a Cauchy function $\psi(x; \mu, \sigma) = (1 + (x - \mu)^2/\sigma^2)^{-1}$.

**Radial filters.** The main component of electrical propagation in neurons consist of a current emission from the center of the cell body to its periphery. A reverse wave sometimes follows. Radial filters are designed to capture these elements. These filters are parametrized by a characteristic radius $\rho > 0$ intended to capture differences in cell body sizes and a propagation velocity $v$ that reflects variations in action potential propagation. This filter family contains two sub-families: emitting filters $v > 0$ and absorbing ones $v < 0$.

**Directional filters.** Ganglion cells favor spatial directions corresponding to their dendritic field on the one hand (site of the input signal), and their axon on the other (output) (see Figure 1). Directional filters capture the propagation velocity $v \geq 0$, direction $\phi \in [0, 2\pi)$ and spatial frequency $\zeta > 0$ of the input and output signals of the cells.

**Circular filters.** Electric currents in cells have rotational motion factors stemming from the shape of their dendritic field (see Figure 1 in the main text), which we parametrize with a phase $\phi \in [0, 2\pi)$, a rotational frequency $\zeta \in \mathbb{N}$ and a velocity $v \geq 0$ component.

Responses to filters are grouped by filter family. Invariance to the phase parameter $\phi$ is achieved by building histograms of filter responses for various values of $\phi$. Further invariance to rotation is encoded through data augmentation by exploiting symmetries of the hexagonal MEA grids as explained in Section 5. Since the relative orientation of the RGC axons and the recording array depends on how the tissue is dissected and lowered on the recording electrodes, it is by no means representative of the cell type and is instead a measurement artifact.

| Filter family | Formula | Parameters | # used |
|---|---|---|---|
| Radial | $\psi\left(r;0,\rho\right)\psi\left(t;\tau,\kappa\right)\cos(r-vt)$ | $\rho,\tau,\kappa,v$ | $2\times 144$ |
| Directional | $\psi\left(r;0,\rho\right)\psi\left(t;\tau,\kappa\right)\cos\left(r\zeta\cos(\theta-\phi)-vt\right)$ | $\rho,\tau,\kappa,v,\zeta,\phi$ | $3\times 192$ |
| Circular | $\psi\left(r;0,\rho\right)\psi\left(t;\tau,\kappa\right)\cos(\zeta\theta-vt+\phi)$ | $\rho,\tau,\kappa,\zeta,v,\phi$ | $2\times 2\times 576$ |

Figure 1: *(Top row)* An example of an Electrophysiological Image (EI) cropped within a $125\mu$m radius of the central electrode over $.5\ ms$. *(Middle row)* Snapshots of a circular filter at different instants in time. *(Bottom row)* Representations of the 3 filter families. The table represents filter expressions in polar coordinates $(r\cos\theta, r\sin\theta, t)$.

## 3  Statistical guarantees on the estimation of y through convex relaxation

Since the estimation of **y** is a matrix completion problem. The standard convex relaxation of this problem can be written using the trace-norm of the variable $\mathbf{Z} \in \mathbb{R}^{n\times n}$ (representing $\mathbf{zz}^\mathsf{T}$ or an arbitrary rank variable introduced to relax the non convex problem), $\|\mathbf{Z}\|_* = \sum_{i=1}^n \sigma_i(\mathbf{Z})$. Here $\sigma_i$ denotes the $i$-th singular value of $\mathbf{Z}$. The convex formulation states that for some choice of the regularization parameter $\gamma > 0$, the minimum of $\frac{1}{2}\|\mathcal{P}(\mathbf{Z}) - \mathbf{c}\|_2^2 + \gamma\|\mathbf{Z}\|_*$ coincides with the minimum of $\ell$. The advantage of this formulation is that the global minimizer of the function is well-characterized. One can use an iterative soft-thresholding algorithm (ISTA [1], see Figure 4 of the main text, middle panel), which, despite slower convergence than the algorithm we proposed in the previous section is guaranteed to solve the problem, and we can also exhibit statistical properties of the minimizer.

We refer the reader to [2] for a general statistical study of trace-norm penalization, and to the appendix for the proof of the following result, which in essence relates the matrix completion-based classification accuracy to the noise amplitude:

**Theorem 1** *Take $\gamma > \|\varepsilon\|_\infty$ such that the minimizer of*

$$\mathcal{L}(\mathbf{Z}) = \frac{1}{2}\|\mathcal{P}(\mathbf{Z}) - \mathbf{c}\|_2^2 + \gamma\|\mathbf{Z}\|_* \tag{1}$$

*has rank one: $\mathbf{Z} = \mathbf{z}\mathbf{z}^\mathsf{T}$. Then the classification accuracy of $sign(\mathbf{z})$, (or $-sign(\mathbf{z})$, whichever is the best) is at least*

$$\mathsf{Accuracy}(sign(\mathbf{z}), \mathbf{y}) \geq \frac{1}{2}\left(1 + \sqrt{1 - \gamma^2 c}\right) \quad with \quad c = 2(\sqrt{2}+1)^2.$$

*In particular if some value of $\|\varepsilon\|_\infty < \gamma < (2-\sqrt{2})/\sqrt{n}$ exists and results in a rank one solution, then this choice of the regularization parameter guarantees exact recovery of $\mathbf{y}$.*

Since the estimation of $\mathbf{y}$ is a matrix completion problem, the standard convex relaxation of this problem can be written using the trace-norm of the lifted variable $\|\mathbf{Z}\|_* = \sum_{i=1}^n \sigma_i(\mathbf{Z})$ where $\sigma_i$ denotes the $i$-th singular value of the matrix $\mathbf{Z}$. The convex formulation states that for some choice of the regularization parameter $\gamma > 0$, the minimum of $\frac{1}{2}\|\mathcal{P}(\mathbf{Z}) - \mathbf{c}\|_2^2 + \gamma\|\mathbf{Z}\|_*$ coincides with the minimum of $\ell$.

We refer the reader to [2] for a general statistical study of trace-norm penalization.

**Theorem 2** *Take $\gamma > \|\varepsilon\|_\infty$ such that the minimizer of*

$$\mathcal{L}(\mathbf{Z}) = \frac{1}{2}\|\mathcal{P}(\mathbf{Z}) - \mathbf{c}\|_2^2 + \gamma\|\mathbf{Z}\|_* \tag{2}$$

*has rank one: $\mathbf{Z} = \mathbf{z}\mathbf{z}^\mathsf{T}$. Then the classification accuracy of $sign(\mathbf{z})$, (or $-sign(\mathbf{z})$, whichever is the best) is at least*

$$\mathsf{Accuracy}(sign(\mathbf{z}), \mathbf{y}) \geq \frac{1}{2}\left(1 + \sqrt{1 - \gamma^2 c}\right) \quad with \quad c = 2(\sqrt{2}+1)^2.$$

*In particular if some value of $\|\varepsilon\|_\infty \leq \gamma < (2-\sqrt{2})/\sqrt{n}$ exists and results in a rank one solution, then this choice of the regularization parameter guarantees exact recovery of $\mathbf{y}$.*

**Proof.** Take $\mathbf{Z} \in \arg\min \mathcal{L}$. First order condition of optimality states that there exists a subgradient $\mathbf{G} \in \partial\|\mathbf{Z}\|_*$ such that

$$0 = \mathcal{P}^*(\mathcal{P}(\mathbf{Z}) - \mathbf{c}) + \gamma\mathbf{G} .$$

Since $\mathbf{c} = \mathcal{P}(\mathbf{y}\mathbf{y}^\mathsf{T}) + \varepsilon$ this writes $0 = \mathcal{P}^*(\mathcal{P}(\mathbf{Z}) - \mathbf{y}\mathbf{y}^\mathsf{T}) - \mathcal{P}^*(\varepsilon) + \gamma\mathbf{G}$. Taking the inner product with $\mathbf{Z} - \mathbf{y}\mathbf{y}^\mathsf{T}$ we get

$$\|\mathcal{P}(\mathbf{Z} - \mathbf{y}\mathbf{y}^\mathsf{T})\|_2^2 = \langle\mathcal{P}^*(\varepsilon), \mathbf{Z} - \mathbf{y}\mathbf{y}^\mathsf{T}\rangle - \gamma\langle\mathbf{G}, \mathbf{Z} - \mathbf{y}\mathbf{y}^\mathsf{T}\rangle . \tag{3}$$

The subdifferential of a convex function is monotonous: taking *any* $\mathbf{G}_0 \in \partial\|\mathbf{y}\mathbf{y}^\mathsf{T}\|_*$ (seen as the trace norm at $\mathbf{y}\mathbf{y}^\mathsf{T}$, not a function of $\mathbf{y}$) we always have $\langle\mathbf{G} - \mathbf{G}_0, \mathbf{Z} - \mathbf{y}\mathbf{y}^\mathsf{T}\rangle \geq 0$. Consequently,

$$\begin{aligned}\|\mathcal{P}(\mathbf{Z} - \mathbf{y}\mathbf{y}^\mathsf{T})\|_2^2 &\leq \langle\mathcal{P}^*(\varepsilon), \mathbf{Z} - \mathbf{y}\mathbf{y}^\mathsf{T}\rangle - \gamma\langle\mathbf{G}_0, \mathbf{Z} - \mathbf{y}\mathbf{y}^\mathsf{T}\rangle \\ &\leq \langle\mathcal{P}^*(\varepsilon), \mathbf{Z} - \mathbf{y}\mathbf{y}^\mathsf{T}\rangle - \gamma\langle\mathbf{y}\mathbf{y}^\mathsf{T}/n + \Pi^\perp(\mathbf{K}), \mathbf{Z} - \mathbf{y}\mathbf{y}^\mathsf{T}\rangle\end{aligned}$$

where $\Pi^\perp(\mathbf{K}) = (\mathbf{I} - \mathbf{y}\mathbf{y}^\mathsf{T}/n)\mathbf{K}(\mathbf{I} - \mathbf{y}\mathbf{y}^\mathsf{T}/n)$ and $\|\mathbf{K}\|_{\mathsf{op}} \leq 1$. Let $\Pi$ denote the orthogonal projection given by $\mathbf{I} - \Pi^\perp$. Taking $\gamma > \|\mathcal{P}^*(\epsilon)\|_{\mathsf{op}} \geq \|\varepsilon\|_\infty$ allows us to choose a matrix $\mathbf{K}$ canceling the noise component in the range of $\Pi^\perp$: $\mathbf{K} = \Pi^\perp(\mathcal{P}^*(\varepsilon))/\gamma$, so

$$\begin{aligned}\|\mathcal{P}(\mathbf{Z} - \mathbf{y}\mathbf{y}^\mathsf{T})\|_2^2 &\leq \langle\Pi(\mathcal{P}^*(\varepsilon)) - \gamma\mathbf{y}\mathbf{y}^\mathsf{T}/n, \mathbf{Z} - \mathbf{y}\mathbf{y}^\mathsf{T}\rangle & (4) \\ &\leq \|\Pi(\mathcal{P}^*(\varepsilon)) - \gamma\mathbf{y}\mathbf{y}^\mathsf{T}/n\|_{\mathsf{F}}\|\mathbf{Z} - \mathbf{y}\mathbf{y}^\mathsf{T}\|_{\mathsf{F}} & \text{(Cauchy-Schwarz)} \\ &\leq (\|\Pi(\mathcal{P}^*(\varepsilon))\|_{\mathsf{F}} + \gamma)\|\mathbf{Z} - \mathbf{y}\mathbf{y}^\mathsf{T}\|_{\mathsf{F}} & \text{(triangle)} \\ &\leq \left(\sqrt{2}\|\Pi(\mathcal{P}^*(\varepsilon))\|_{\mathsf{op}} + \gamma\right)\|\mathbf{Z} - \mathbf{y}\mathbf{y}^\mathsf{T}\|_{\mathsf{F}} & \text{(dim range } \Pi = 2) \\ &\leq (\sqrt{2}+1)\gamma\|\mathbf{Z} - \mathbf{y}\mathbf{y}^\mathsf{T}\|_{\mathsf{F}} & \text{(assumption } \gamma \geq \|\mathcal{P}^*(\varepsilon)\|_{\mathsf{op}}) .\end{aligned}$$

On the other hand we know that if $\gamma$ is taken large enough so that $\mathrm{rank}(\mathbf{Z}) = 1$, then writing $\mathbf{Z} = \mathbf{z}\mathbf{z}^{\mathsf{T}}$, and using the fact that the sparsity pattern of $\mathcal{P}$ contains the main diagonal, we get the sequence of inequalities

$$
\begin{aligned}
\|\mathcal{P}(\mathbf{Z} - \mathbf{y}\mathbf{y}^{\mathsf{T}})\|_2^2 &\geq \sum_{i=1}^{n}(\mathbf{z}_i^2 - 1)^2 \\
&= \|\mathbf{z}\|_4^4 - 2\|\mathbf{z}\|_2^2 + n \\
&\geq \|\mathbf{z}\|_2^4/n - 2\|\mathbf{z}\|_2^2 + n \qquad \text{(Cauchy-Schwarz)} \\
&= \frac{1}{n}(\|\mathbf{z}\|_2^2 - n)^2 = \frac{1}{n}\|\mathbf{Z} - \mathbf{y}\mathbf{y}^{\mathsf{T}}\|_{\mathsf{F}}^2
\end{aligned}
$$

and therefore $\|\mathbf{Z} - \mathbf{y}\mathbf{y}^{\mathsf{T}}\|_{\mathsf{F}} \leq (1 + \sqrt{2})\gamma n$.

Note that flipping signs of the same entries of $\mathbf{z}$ and $\mathbf{y}$ does not affect the Frobenius norm of the difference $\mathbf{z}\mathbf{z}^{\mathsf{T}} - \mathbf{y}\mathbf{y}^{\mathsf{T}}$:

$$
\forall \mathbf{s} \in \{1, -1\}^n \quad , \quad \|\mathbf{z}\mathbf{z}^{\mathsf{T}} - \mathbf{y}\mathbf{y}^{\mathsf{T}}\|_{\mathsf{F}}^2 = \|\mathrm{Diag}(\mathbf{s})(\mathbf{z}\mathbf{z}^{\mathsf{T}} - \mathbf{y}\mathbf{y}^{\mathsf{T}})\mathrm{Diag}(\mathbf{s})\|_{\mathsf{F}}^2
$$

Therefore consider for simplicity, and without loss of generality, that $\mathbf{y} = \mathbf{1}$. Let $q$ denote the number of negative elements of $\mathbf{z}$, and with the convention $\mathrm{sign}(0) = 1$, we know that $\mathbf{z}\mathbf{z}^{\mathsf{T}}$ has $2q(n-q)$ negative elements. Consequently we have

$$
\|\mathbf{z}\mathbf{z}^{\mathsf{T}} - \mathbf{1}\mathbf{1}^{\mathsf{T}}\|_{\mathsf{F}}^2 \geq 2q(n-q) \ . \tag{5}
$$

Let $\xi = q/n$ denote the ratio of misclassified elements. We have, combining Eqs. (5) and (4),

$$
2\xi(1 - \xi) \leq (1 + \sqrt{2})^2 \gamma^2 \ ,
$$

and therefore

$$
\xi \leq \frac{1}{2} - \frac{\sqrt{1 - 2\gamma^2(\sqrt{2} + 1)^2}}{2} \ . \quad \square
$$