[Reviews · NeurIPS 2015]

Submitted by Assigned_Reviewer_1

I think this paper is going to be a forerunner to a journal paper (as it should be), and this paper outlines the methods they use.

I think the authors spent too much time describing the features (which seemed intuitive to use); I would have preferred they put that extra space to giving more intuition/clarity to their methods (and defining variables better).

In the end, it would have been nice to say something about the unlabelled data (i.e., our classifier identified this number of midget cells, etc., which is consistent with previous reports).

Notes while reading:

015 - "real neural networks" I would just call "populations of neurons"; neural networks is already a buzzword

050 - I think the BMI argument is weak-this work has nothing to do with BMI.

I like the first point much better, and you can probably extend it...it is now possible to record from a complete population of neurons in the retina, and by identifying all cell types, we can have a complete picture of this population (something along those lines).

Right now, I think it is too lofty to assume we could directly stimulate each neuron in the population independently.

088 - what is n and what is p?

119 - I feel like there's a bit of notation abuse (i.e., too many variables that are not well defined; for the reader, it's tough to keep track 1) what the variable means and 2) why do you need it)

188 - typo for "electrophyiological"

Figure 1: you'll need to do a lot better job explaining this figure..what does red dot > 0 mean?

what does each line in the distance to soma mean?

I had a really hard time, even looking back, at what this figure is trying to convey.

212 - I'm losing track here.

30 x 19... you are looking at 19 electrodes in 125us radius, and the 30 is from the number of time points you are sampling from the voltage waveform?

Figure 2- what are the red lines?

averages? each purple line is a pairwise cross-correlation function?

259 - "ganglion cells within the same functional class" needs a reference also next sentence

314 "4rd" typo 314 I don't understand what you mean by the 4th and 5th columns.

why would you use EIs only and ISIs only (do you mean green and blue lines?)

355 - you should not have the same colors for figure 3 if they mean different things in the right panel

Figure 4 left panel: label colormap legend

395 - don't use hyphens to emphasize "-hard-"

427 - is it clear that different cell types always have different functional properties?

looking back at the title, it's not very descriptive of the work
Summary: I liked the idea of the paper, and it's probably on the better end of papers with neuroscience focus.

However, I felt the paper layout needs work---the reader can get lost quite easily, as the text needs to provide more intuition.

Submitted by Assigned_Reviewer_2

The paper is well written, and considering the number of results presented, fairly clear.

To my knowledge, the author's approach to the problem is quite original.

The use of methods like this may be quite important for large scale neuroscience. With that said, what follows are a list of more minor comments, which may improve the clarity of the paper:

The paper could be improved by moving non-essential results to the appendix.

For example, the theory regarding the convex relaxation of the matrix completion algorithm used to identify cell polarity seems non-essential to the main body of the paper.

The description of data slices is somewhat confusing.

When describing the matrix, X, on line 88, it may be helpful to give concrete meanings to the dimensions (I believe n to the the number of data points and p to be the dimensionality of a slice).

In the right panel of figure 3, it is not clear what the dashed lines mean.

The text says these "correspond to predictions by the first order (nonconvex) method stopped after 100 steps."

This is then different than the Newton method presented in the paper.

Is this first order method ever described?

After reading the paper, I was not completely clear on how matrix completion is used to improve cell polarity estimates.

It seems that the logistic regression is capable of producing an estimated polarity for each cell.

When using matrix completion to predict polarity, is this information discarded, and only the coupling strengths used or is logistic regression and matrix completion used in some two-step process to predict polarity?

For readers who are more used to recording with electrode arrays in cortex (vs. retina) the array used here may be somewhat novel.

It may be helpful to make explicitly clear that unlike cortex, the voltage activity of a single neuron is picked up on multiple electrodes.

Summary: The authors consider a novel approach to using machine learning techniques to automatically identify cell types in the retina.

The authors apply a suite of sophisticated methods to the problem, include theory for a matrix completion problem motivated by their scenario, empirically investigate different methods for solving the non-convex problems that arise in their scenario and ultimately show impressive classification results.

After further review, I continue to feel the author's have done considerable work in an interesting way, but I have lowered my score to reflect the lack of clarity in the current presentation of the paper.

Submitted by Assigned_Reviewer_3

The authors propose to identify different types of retinal ganglion cells based on their electrical image in high-density MEA recordings, their inter-spike interval distribution and their couplings to other cells. To this end, the authors focus on the problem of identifying suitable features from these measurements to then train logistic regression or other classifiers for inferring the cell type. In addition, they use matrix completion techniques for inferring the cell polarity (ON vs OFF). Identifying retinal ganglion cell types is clearly a timely topic (reviewed recently by Sanes and Masland, Annual Reviews of Neuroscience, 2015) and the approach chosen by the authors is creative and interesting: typical techniques either focus on anatomical reconstructions or functional responses.

However, I found the the paper was not well structured, so that it was difficult to assess precisely what was done and what are the novel contributions.

The first contribution was to use certain features and apply a logistic regression (or other) classifier. The features used for the classification algorithm in lines 79-101 are not well described. The authors should describe the manually chosen filter banks in the main paper, not only define them in the appendix. They should also define what slices are and how they look, how the filter banks were chosen and what the clusters were. It seems to me that much of the information later presented in section 4 should have already been worked into section 2.

The second contribution is to estimate a "polarity" vector by exploiting the cross-correlation matrix and the fact that types of same polarity have positive correlations. It is a bit unclear how this task relates to the previous one. They describe their approach to this "matrix completion" problem in some detail on page 3, which again is very dense and full of jargon (e.g. what is a lifted matrix?). Theorem 1 seems oddly out of place - while this may be an interesting theoretical result, a data analysis paper seems to me not the place to advertise it. Section 5.2 seems of place after the results and I am not sure I understand its relevance (partially because Fig. 4 is hard to read and lacks meaningful labels).

The evaluation of the methods seems mostly performed with

care and using interesting setups, e.g. the transfer between datasets. Unfortunately, the hand labeled data only includes the standard set of ON/OFF midget and parasol RGCs, so the evaluation is restricted to those types. In some sense, those are the easy ones, as they are so abundant in the data and can be clearly identified in primate recordings. Of course, for testing a new method that may be fine. The data augementation for building in invariances is a nice trick, albeit not new.

The results seem are ok, but it is hard to tell what a good baseline would be to compare against - on the task of polarity identification, the method devised by the authors only performs a little better than just exploiting ISI/CCs. Indeed, it is unclear what the values in () in Table 1 are, so it may be that the author's method does not significantly outperforms the simpler methods. In line 351ff, the authors write that performance was >90% on 3 datasets for polarity inference, and ~66% for the rest. Are those included in the numbers reported in the table and the figures? What is the difference between the datasets? Then they say they only show the results for 4/10 datasets? Which ones? Regarding type identification, features from the electrical image clearly improve it, although random features seem to be as good as handcrafted ones (~93% correct). What about PCA features of the electrical image (or other standard feature sets one might think of)? In Fig 2 right, I am somewhat puzzled by the red curve. Wouldn't you expect that this should be high where green and blue curves are high? Can the authors provide any insight into why the red curve is only high for smaller distances, although the task is to infer polarity as for the green curve? In Fig. 3, error bars are again unclear. An analysis of the false positive/false negative rates would have been nice to see.

Summary: This approach has promise, but needs a longer and more accessible exposition and more analysis. Concentrating on fewer aspects would have been helpful.

Submitted by Assigned_Reviewer_4

The authors present a method for classifying retinal ganglion cells based on their electrical signatures on an electrode array, attempting to identify three cell types and their polarities if applicable. I am no expert in this field but feel that the authors made an objective and informative report of their methods and results without drawing undue conclusions. A question I cannot answer--that the authors may have wanted to address--is to what extent the current results are indeed good enough to

make e.g. automatic mining practically useful, but it seems clear that a noteworthy step has been made.

The paper is well-written and clearly structured, and is relevant to the NIPS community.
Summary: Interesting work attempting to automatically identify both type and polarity of retinal ganglion cells using a classifier trained and applied on recorded voltage patterns. The authors report their results and propose next steps.

Author Feedback
Author rebuttal: We thank reviewers for a careful and critical read of the manuscript.
R-2
Identifying cell-types from electrophysiologiocal recordings is crucial in neuroscience. We describe a new strategy. We separated algorithms (Sec 2&3) from practical implementation to emphasize on the algorithms that the wider neuroscience community might be interested in, before specifically focusing on RGCs.   

l. 79-101. for each slice we filter the EI with 1 family of filters (see appendix for closed-form expressions). Different filters can be used. Due to the length constraint, we only compare random projections with a hand-built filter bank (Appendix). Eq. (1) is minimized, where X is a slice of unlabeled data to get V.  Then for each labeled data point we soft-threshold V'x , see l. 96-97 [3,4,15].

Matrix completion. The lifted matrix Z represents z z' or a variable of larger rank that we use to relax the nonconvex objective. We will add this. Th 1 provides us with an understanding of what factor in the input data is responsible for the variability of the results we empirically experienced. It tells us that the noise maximum entry is the determining quantity for accuracy. To the best of our knowledge, this is the first time matrix completion is used for binary classification, making this result interesting by itself.

CCF data: value at the origin is compared to the average: the difference (value at 0) - (average) is what we feed to the matrix to be completed. This is explained l. 268-269.  We added an equation to remove ambiguity.

Values in tab 1. CCFs predict only polarities, the average is reported for 2 reasons: a) the idea of using matrix completion here is new b) it has a high potential, as demonstrated by fig 3, right hand panel. On some datasets, matrix completion has 100% accuracy. However, on other datasets, either because of issues a fragile spike-sorting, or of other noise, the approach does not do as well. Th 1 helps understand this phenomenon.

For visibility we report results on 4 datasets: the best, the worse and 2 intermediary. We aimed at showing the potential of the method and its weaknesses: in particular sensitivity to maximum cell distance is clear.

Sec 5.2. Our heuristic for initializing kmeans showed surprisingly powerful. Proving theoretical statements about this algorithm is technically very difficult due to nonconvexity and the nontrivial interaction of variables. While we could not address the problem in the scope of this work, our coarse-to-fine optimization strategy was successful/innovative enough to be worth reporting. Fig 4 (right) reports values of Eq. (1) as a function of iterations. 5 plots corresponding to different initializations reported in the legend.

We showed how to leverage the large amount of data available from Midget/Parasol RGCs, and our method could easily be extended to other cell-types if abundant recordings were available. We find the comment "results are not stellar" rather harsh. Our data will be released to make comparisons possible.

R-7
BMI. In the retina, reproducing the neural code with an artificial device in blind people will require identifying cell types so that appropriate signals can be delivered.  Cell type identification cannot be done with a) light responses: no light responses (blind) b) anatomical methods: not yet usable in a living eye.  However, our method could identify distinct cell types using a retinal prosthesis.   In the short term we do not envision these algorithms running in the BMI device itself, instead, we envision a calibration of the device for the patient in a clinic, and the creation of a detailed "addressing map" indicating which electrodes in the device activate which cell types. This approach and the need for the algorithms in our paper discussed in Jepson et al 2015. Text edited for clarity.

n = # observations p = # descriptors per slice. We will add a note. 30 is the number of time samples.

Fig 1: we meant to convey some intuition as to how our data looks like to the machine-learners. Legend is clarified. The top row illustrates the physical principles that lead to EI: a multielectrode array records a voltage signature that corresponds to a downsampled projection of the morphology of the cell onto the place of the recording array.

4&5th cols tab 1: EIs & ISIs used as a comparison baseline to show they bring orthogonal information.

R-3

These results are a proof of concept. Variability across datasets is currently too high to rely on our classifiers for scientific or clinical use, however, they show promise in being able to mine large data sets, which is impractical with manual approaches.  The approach could also form the basis for improvements to future clinical devices. We aim to train classifiers on larger datasets in the future, the bottleneck being the hard manual work required to acquire high quality labelled data for verification. Will emphasize this in conclusion.